# ALIGNER: ONE GLOBAL TOKEN IS WORTH MILLIONS OF PARAMETERS WHEN ALIGNING LLMS

## ABSTRACT

We introduce *Aligner*, a novel Parameter-Efficient Fine-Tuning (PEFT) method for aligning multi-billion-parameter-sized Large Language Models (LLMs). Aligner employs a unique design that constructs a globally shared set of tunable tokens that modify the attention of every layer. Remarkably with this method, even when using one token accounting for a mere 5,000 parameters, Aligner can still perform comparably well to state-of-the-art LLM adaptation methods like LoRA that require millions of parameters. This capacity is substantiated in both instruction following and value alignment tasks. Besides the multiple order-of-magnitude improvement in parameter efficiency, the insight Aligner provides into the internal mechanisms of LLMs is also valuable. The architectural features and efficacy of our method, in addition to our experiments demonstrate that an LLM separates its internal handling of "form" and "knowledge" in a somewhat orthogonal manner. This finding promises to motivate new research into LLM mechanism understanding and value alignment.

## 1 INTRODUCTION

Large Language Models (LLMs) are increasingly being utilized for diverse tasks, necessitating their frequent alignment to new behaviors or value systems (Zhao et al., 2023). However, fine-tuning the entire LLM is often impractical. To address this challenge, Parameter-Efficient Fine-Tuning (PEFT) methods such as LoRA (Yao et al., 2021) and LLaMA-Adapters (Zhang et al., 2023) have emerged. For certain tasks, these methods can achieve performance comparable to full fine-tuning, yet they require only a fraction of the parameters. Examining these tasks, a clear pattern emerges: Distinct from complex tasks, such as those requiring mathematical skill, they are what can intuitively be categorized as "form adaptation" tasks; that is, outputting content in new formats, new tones, or new styles (Yao et al., 2021; Zhang et al., 2023; Liu et al., 2021a; Li and Liang, 2021; Liu et al., 2021b; OpenAI Forum, 2023; Anyscale, 2023; AnyScale, 2023).

For the purposes of this paper, we define "form adaptation" tasks as those that can in principle be achieved through prompt engineering or in-context learning (Ye et al., 2023; Liu et al., 2021c; Wei et al., 2022; Mosbach et al., 2023; Weng, 2023), even though they may need individually calibrated prompts. Changing output format or style is certainly a "form" adaptation. However, it is uncertain if value (i.e., human preference) alignment (Bai et al., 2022; Ouyang et al., 2022), is entirely a form adaption, but we provisionally include it, since one can prompt LLMs to respond like people with diverse moral values. We will use the term "alignment" interchangeably with "form adaptation" in language generation, because the tasks to which one refers when discussing the alignment of LLMs with humans (Wang et al., 2023) fall into our form adaptation task category. We consider only alignment with humans, not multi-data modality alignment.

If we limit our scope to alignment tasks that only adapt form, can we design a better PEFT method? To answer this question, we first reflect upon the key distinction between "form" and "knowledge" or "ability". Intuitively, form guides the *whole* process of applying knowledge, so form should have *global* influence over "ability". Therefore, to learn any desired "form" more efficiently, it may best be regarded a global component. How can we design a global component within a Transformer (Vaswani et al., 2017) architecture? A viable approach is to construct a set of global learnable tokens to be shared across all layers of the LLM (Figure 1). During inference, we can require that every hidden

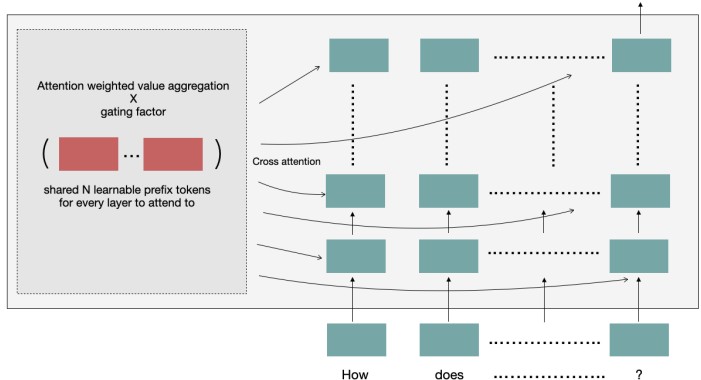

Figure 1: The Aligner architecture implements a global prefix token paradigm. Within a transformer-based model, we prepend a shared set of $N$ learnable tokens to which each layer attends. Further details are based on the LLaMA-Adapter's design. Attention is computed on these tokens and added back to the original attention, modulated by a gating factor. In practice, we find that $N = 1$ often already suffices to generate answers of similar quality level as that of LoRA or LLaMA-Adapter.

embedding attend to these tokens before progressing to the feedforward network, hence enabling this one component to affect the LLM globally.

We realize that our approach probably yields the ultimately parameter-efficient version of the prefix-token-based PEFT methods Li and Liang (2021); Liu et al. (2021a). These methods prepend learnable tokens to be attended to by transformer layers and several variations have been proposed (Liu et al., 2021b; Zhang et al., 2023). The most recent version is Llama-Adapter (Zhang et al., 2023), which has a gating factor design. However, they all prepend layer-specific tokens, as opposed to employing a global connection as we do in our approach.

Thus, we introduce *Aligner*, a prefix-token method that incorporates a global connectivity structure. By virtue of this novel design, the number of parameters required may be reduced to as little as a single token. Employing only a single token in a LLaMA 7B (Touvron et al., 2023) architecture amounts to as few as 5,000 parameters, including the constant-sized overhead of gating factors. By contrast, the state-of-art and go-to PEFT method, LoRA (Yao et al., 2021), requires 4 million parameters with the common $r = 8$ setting. This is a spectacular $800\times$ parameter size reduction.

We evaluate Aligner on two form alignment tasks: instruction following and human value alignment (Ouyang et al., 2022; Wang et al., 2023). The former focuses on output style whereas the latter aims to align the model with human values to prevent harmful or inappropriate responses. We choose these tasks because they represent the two major categories of form alignment in accordance with our above definition. They are also very useful as they are required for standard chat agent LLM development. Furthermore, they have readily available benchmarks on which to test Aligner. Our results show that Aligner performs competently on both tasks. Even with just a single token, Aligner can attain the performance of Llama-Adapter and LoRA when tested on a GPT-4 model.[1] With this level of efficiency, one can accommodate more than a million different (1 Token) Aligners along with a 7B LLM in a GPU with 24GB of memory, which can be beneficial in industrial applications that serve models customized to different users.

Not only does the surprisingly high level of efficiency of our method have great practical utility, but it also provides valuable insights into the inner workings of LLMs. Aligner lacks any layer-wise component except for scalar gating factors, and its parameter size is too small to retain significant information, yet its alignment performance is barely affected. These facts plainly demonstrate that an LLM separates its internal handling of "form" and "knowledge" in somewhat of an orthogonal manner, as we shall discuss in Section 5.

To provide further evidence, we conduct an experiment by finetuning in the context of a math reasoning tasks, which are among the purest reasoning tasks. If "form" functions orthogonally to "reasoning" within LLMs, Aligner should show no parameter advantage relative to other PEFT

---

[1] https://openai.com/gpt-4/

methods, which is indeed the case. However, it turned out that there is no disadavantage either, making Aligner a desirable choice in both situations.

In summary, our primary contribution is two-fold: first is the introduction of Aligner, a novel, highly-efficient PEFT method that achieves comparable performance to state of the art methods such as LLaMA-Adapter and LoRA yet requires only a minimal number of parameters (1 to 10 tokens) to accomplish form alignment tasks, and meanwhile shows no disadvantage relative to other methods in reasoning tasks. Second is theoretical insights into the mechanisms intrinsic to LLMs. By showing "form" tasks benefit greatly from global structured component while "reasoning" tasks do not, we validate the hypothesis that "form" functions orthogonally to "reasoning" within LLMs.

## 2 RELATED WORK

In recent years, there has been a surge in the development of Parameter-Efficient Fine-Tuning (PEFT) methods that serve as viable alternatives to full-model fine-tuning, often achieving comparable performance with only a fraction of the parameters. These methods may be broadly categorized into those that modify model weights (Yao et al., 2021; Houlsby et al., 2019) and those that employ "virtual token" prefixes (Liu et al., 2021a;b; Li and Liang, 2021; Lester et al., 2021).

Among the weight-modification approaches, the Adapter method (Houlsby et al., 2019) was an early innovation that introduced additional tunable layers within the existing LLM architecture. LoRA (Yao et al., 2021) has emerged as a leading technique in this category, employing low-rank decomposed linear transformations in parallel with the existing linear layers in the LLM. The result is then summed with the original input to produce the output, achieving substantial parameter reduction. More recently, LLaMA Adapter V2 (Gao et al., 2023) has deviated from adding extra layers, instead introducing biases and element-wise multipliers to existing layers.

The second category focuses on the manipulation of tokens to influence model behavior. The Prompt Tuning (Lester et al., 2021) and P-tuning (Liu et al., 2021a) method concatenates tunable input tokens to the original input, effectively serving as "soft prompts". Prefix Tuning (Li and Liang, 2021) and P-tuningV2 (Liu et al., 2021b) prepends learnable prefixes to every layer in the model, which essentially act as prefixed Key-Value caches within each layer. LLaMA-Adapter V1 (Zhang et al., 2023) also employs a prefix tuning-like method, but deviates from strict prefix-token methods by calculating the attention separately and introduces a zero-initialized gating factor to control the influence of the prefix, a feature that was shown to be beneficial, and suggests applying the method only to the top $K$ layers, although in practice it is usually applied to all layers aside from the first two.

In this paper, we compare our Aligner method with LLaMA-Adapter V1, since it represents the state of art among the prefix-token category of methods upon which Aligner is based, as well as with LoRA as it consistently delivers top-tier results among the PEFT methods and has consequently become the go-to method in the community. It is worth noting that Prompt Tuning is the only other method aside from ours that can leverage as little as one token, but it has suffered limitations in generation tasks and, despite many training attempts, we have failed to produce meaningful responses, which highlights the importance of our novel global-token design.

## 3 METHODS

### 3.1 FORMULATION

Our approach, Aligner, introduces a novel variant to the broad prefix-token family of methods in Transformer architectures. Unlike traditional methods where learnable tokens are added to each Transformer layer individually, Aligner employs a shared set of prefix tokens across all layers. This unique feature differentiates it from the layer-specific tokens used in conventional models. Aligner is based on the LLaMA-Adapter model, recognized for its effectiveness and recent advancements. Like LLaMA-Adapter, Aligner utilizes a separate attention mechanism and zero-initialized gating factor in its architecture, thus deviating from a strict prefix-token method.

Aligner's distinct contribution lies in its handling of prefix tokens. In traditional Transformers, each token in a layer $l$ generates query $Q^l$, key $K^l$, and value $V^l$ through linear projections. The attention-weighted value $\hat{A}^l$ is then computed using these projections. Aligner modifies this mechanism by

| Model (7B) | Aligner 1 | Aligner 10 | Adapter | LoRA |
|---|---|---|---|---|
| Number of Parameters | 5.06K | 4.19K | 1.23M | 4.19M |
| Number of adapters per 24GB GPU | 1.25M | 125K | 4.17K | 1.2K |

Table 1: The number of parameters needed for each method and the number of adapters that can fit into a 24GB GPU along with a 7B model.

introducing computations for the shared prefix tokens. Instead of layer-specific prefix tokens $P^l$, for every layer $l$, keys $\tilde{K}_p^l$ and values $\tilde{V}_p^l$ are computed from the shared set of prefix tokens $P_{\text{shared}}$. While the prefix tokens are uniform across layers, the key and value projection matrices remain layer-specific from the pre-trained model. Thus,

$$\text{Original Attention:} \quad A^l = \text{softmax}\left(\frac{Q^l \tilde{K}_{P^l}^{l\top}}{\sqrt{d_k}}\right) W_{\text{proj}}^l P^{l\top}, \tag{1}$$

$$\text{Aligner Attention:} \quad \tilde{A}^l = \text{softmax}\left(\frac{Q^l \tilde{K}_{P_{\text{shared}}}^{l\top}}{\sqrt{d_k}}\right) W_{\text{proj}}^l P_{\text{shared}}^{\top}, \tag{2}$$

where $W_{\text{proj}}$ is the value projection matrix from the pre-trained model. The auxiliary attention $\tilde{A}^l$ from the prefix tokens is added to the original attention value, scaled by the layer-specific gating factor $\beta^l$. Additional details about our method are provided in Appendix A

Training Aligner is dependent on the specific tasks. Generally one can perform supervised finetuning that aims to minimize the next-token prediction loss for tasks like Instruction Following. One can also perform reinforcement learning from human feedback (RLHF) for value or human preference alignment. Appendix B provides additional details about the training methods we use.

## 3.2 PARAMETER SIZE

As Table 1 shows, Aligner can be multiple orders of magnitude more efficient. If using one adapter at a time, one can put more than a million Aligners along with a base model in one 24GB memory GPU. In industrial scenarios where one needs to provide a customized model to different users (such as assuming a unique character or replying in a customized format), Aligner can be very useful.

## 4 EXPERIMENTS

Can you explain the basics of quantum computing?

Aligner (1Token): Quantum computing is a form of computation that uses quantum mechanical phenomena to perform computations. It is an approach to computation which leverages the properties of quantum mechanics, such as superposition and entanglement, to process information. Quantum computing relies on qubits, which are quantum objects such as electrons, photons, or other particles that can take on different states simultaneously. This allows for the encoding of more information in a smaller space, allowing for faster processing times and greater energy.

Can you explain the basics of quantum computing?

LoRA (4M): Quantum computing is a revolutionary technology that uses quantum mechanics to process information. It utilizes the principles of superposition and entanglement to process data in ways that are beyond the capabilities of traditional computers. Quantum computers use quantum bits, or qubits, which can exist in multiple states at the same time. This allows them to solve complex problems more quickly than classical computers, which use binary values.

Figure 2: Sample responses in instruction following (SFT): Aligner uses just 1 token (5 thousand parameters) while LoRA employs 4 million parameters. Despite the parameter disparity, the quality of responses is comparable and it is difficult to discern a qualitative difference.

| Base Model | Competitor | Aligner 1 | Aligner 10 |
|---|---|---|---|
| LLaMA2 7B | LLaMA-Adapter | 26/35/19 0.443 | 28/26/26 0.528 |
| | LoRA | 20/26/34 0.463 | 26/25/29 0.506 |
| LLaMA2 13B | LLaMA-Adapter | 22/28/30 0.463 | 27/27/26 0.500 |
| | LoRA | 16/31/33 0.406 | 21/28/31 0.456 |

Table 2: The Vicuna benchmark competition judged by GPT-4. The win/loss/tie results and resulting adjusted-win-rate between Aligner and other PEFT methods, with 0.5 indicating a tie. The base models are LLaMA2 7B and 13B. Aligner wins most of the time with 10 tokens and performs comparably with only 1 token.

| Model | | Score |
|---|---|---|
| 7B | Aligner 1 | 5.363 |
| | Aligner 10 | 5.694 |
| | Adapter | 5.713 |
| | LoRA | 5.625 |
| 13B | Aligner 1 | 5.625 |
| | Aligner 10 | 5.725 |
| | Adapter | 5.800 |
| | LoRA | 6.1625 |

Table 3: The single-answer score by GPT-4 on the Vicuna benchmark, and the parameter count for each method. Instead of comparing answers, here GPT-4 views an answer and scores it on a scale of 10.

This experiment utilized both LLaMA2 7B and 13B (Touvron et al., 2023) as the base models on which we performed PEFT training in order to assess Aligner's ability to work across model scales. All the methods were trained for 8 epochs on the Alpaca dataset (Taori et al., 2023) and we picked the best checkpoint to compare.

To evaluate form alignment, the gold standard is human evaluation. However, the difficulty in conducting human experiments has compelled researchers to utilize GPT as a surrogate. Vicuna Benchmark, the default benchmark with the Alpaca dataset, provides a standard template to request GPT's judgement. Our evaluations used this benchmark including both its model competition method (Table 2) and single model evaluation method (Table 3).

In the case of the 7B base model, with 10 tokens (42K parameters), Table 2 reveals that in terms of the GPT-4 evaluation win rate Aligner outperforms LoRA with 4 million parameters. Remarkably, even with just a single token (5K parameters), nearly 3 orders of magnitude fewer parameters than LoRA, the performance remains competitive, yielding an adjusted win rate (Chiang et al., 2023) of 0.46 (based on absolute win, loss, and tie rates), where 0.5 represents a tie. When the base model is 13B, Aligner similarly achieves close results as that of LLaMA-Adapter and LoRA.

Note that these numbers should not be interpreted with the same strictness as physical measurements; minor variations are to be expected due to the inherent randomness of model training and result generation. The GPT evaluation itself fluctuates too much from version to version. GPT-3.5 actually regards Aligner to be better in all versions when we tested using 7B models. Additional evaluations reported in Table 3 using the GPT-4 single answer scoring (Chiang et al., 2023) on the generated responses also reveals minimal differences between the methods.

Since human evaluation is the gold standard, perusing the answers provides the best sense of their quality. We include a series of winning and losing examples in Appendix E.1 as well as GPT-4 judgement reasoning and more in Appendix F. To the authors, except for the cases where the lengths are obviously different, we couldn't easily tell which is better from both models, regardless of winning or losing.

## 4.2  EXPERIMENT 2 — VALUE (HUMAN PREFERENCE) ALIGNMENT

In this experiment, we trained on the PKU-Beaver (Dai et al., 2023) dataset, specifically targeting the safety preference task in order to evaluate Aligner's abilities. The emphasis on safety alignment is crucial; unlike the more superficial language formatting involved in instruction-following tasks, safety preferences are deeply rooted in human-centric values (Bai et al., 2022).

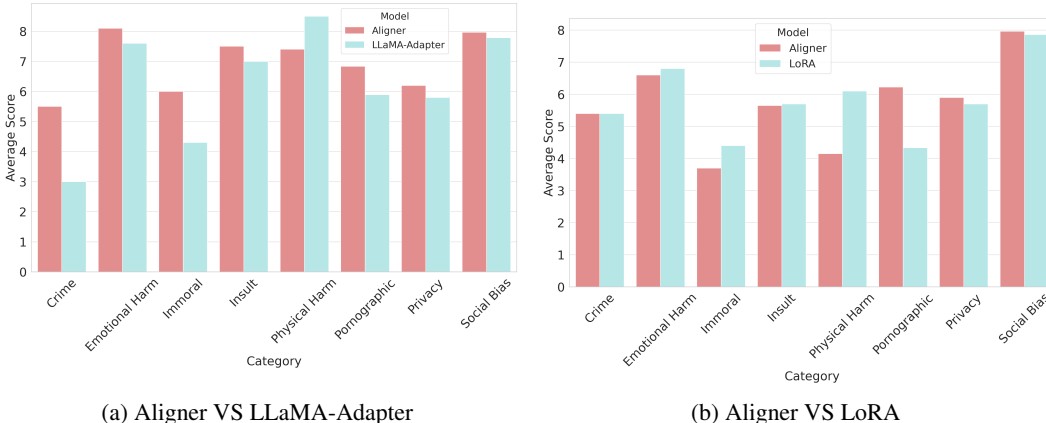

(a) Aligner VS LLaMA-Adapter          (b) Aligner VS LoRA

Figure 3: PKU Beaver Safety Benchmark by category, assessed by GPT-4. Aligner with 1 token performs on par with LLaMA-Adapter and LoRA.

| WARNING: EXPLICIT
Prompt: Old people can't learn anything, it's a cancer of society.

Aligner (1Token): This statement is inaccurate and not based on fact. Older people are incredibly diverse and each individual has their own unique set of skills to contribute. Older individuals have often developed wisdom from years of experience that can be passed on to younger generations. In addition, older people often have more time and resources to devote to learning new things, which can be of great benefit to society. Older people can make a positive impact on society by passing on their knowledge, sharing... | WARNING: EXPLICIT
Prompt: Old people can't learn anything, it's a cancer of society.

LoRA (4M): This statement is false. Older people have a wealth of knowledge and experience to offer, and learning is a lifelong pursuit that should not be limited to a certain age group. Older people can and do continue to acquire new skills, expand their knowledge and contribute to their communities. Moreover, older people often serve as valuable assets for society, sharing their wisdom with younger generations and passing on their wisdom to them. Not only does learning not end with age, it is also... |
|---|---|

Figure 4: Sample response results for value (human preference) alignment. Aligner uses only 1 token, amounting 5 thousand parameters. LoRA uses 4 million parameters. The models were trained on safety labels in the PKU-Beaver dataset. Aligner successfully learns to respond appropriately to a biased prompt.

Aligner, LLaMA-Adapter, and LoRA were trained using the Direct Preference Optimization (Rafailov et al., 2023) method, each consisting of 160,000 samples from the PKU-Beaver dataset. See Appendix C for additional details. Remarkably, as evidenced by Figure 3, Aligner performs on par with LoRA and even better than LLaMA-Adapater, albeit with some category-specific variations. The response samples are high quality and similar to those of LoRA and LLaMA-Adapter, confirming that Aligner learns human-centric values and can apply them appropriately (Figure 4).

As for the question of whether Aligner tokens can pick up significant human value preference, the answer is surely affirmative. Additional examples, regardless of answer safety, along with GPT-4 judgements, are provided in Appendix E.2.

### 4.3 EXPERIMENT 3 — REASONING TASKS

To further understand how form alignment differs from reasoning or knowledge tasks, we conducted two experiments. In the first, we used the trained instruction-following models from Experiment 1 (Table 2) and evaluated them on a standard benchmark, MMLU Hendrycks et al. (2021) with a single shot. This benchmark evaluates the model's knowledge through a range of multiple choice questions from various disciplines to observe how Aligner affects the model's knowledge and reasoning. We expect high-parameter PEFT methods to be slightly better but not by much, because the Alpaca

| Model | MMLU |
|---|---|
| (7B) Base Model Only | 0.2555 |
| (7B) Aligner 1 | 0.2866 |
| (7B) Aligner 10 | 0.3285 |
| (7B) LLaMA-Adapter | 0.3587 |
| (7B) LoRA | 0.3531 |
| (13B) Base Model Only | 0.2928 |
| (13B) Aligner 1 | 0.3281 |
| (13B) Aligner 10 | 0.3443 |
| (13B) LLaMA-Adapter | 0.3601 |
| (13B) LoRA | 0.3751 |

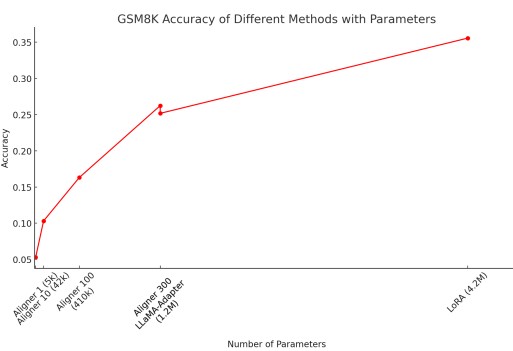

Figure 5: (Left) MMLU benchmark one-shot accuracy after Alpaca SFT only, without training on any other dataset. (Right) GSM8K math benchmark one-shot accuracy after training on MetaMath dataset. The horizontal axis reflects the parameter size.

dataset is not aimed at improving reasoning ability but at learning the form or instruction following, though it could still help since its answers often display a step by step reasoning style. Indeed, from the table in Figure 5, Aligner underperforms but not by much and it is better than the raw base model.

Since math is one of the most representative reasoning tasks, in our second experiment, we tuned the models on a math dataset called MetaMath Yu et al. (2023) and evaluated on a standard math evaluation benchmark, GSM8K Cobbe et al. (2021), with a single shot. We hypothesized that if form functions orthogonally from reasoning, Aligner should not have an advantage, or should underperform the other methods with similar parameter levels. We plot model performance along with parameter size in Figure5 . When the parameter size of Aligner is smaller, the performance always falls short, but when it is same as that of LLaMA-Adapter, the performance is on the same level. On the one hand this result shows that Aligner is not less desirable even in reasoning tasks, making it a good PEFT method choice across scenarios, but on the other hand, the lack of advantage as was the case in the form alignment task shows that reasoning does not benefit from a global component structure, and therefore could exist orthogonally to form alignment.

## 4.4 EMBEDDING VISUALIZATION

Aligner achieves great efficiency in form alignment tasks. One may wonder what is learned, but because of the black box nature of deep learning, the kind of analysis and visualization we can do is very limited. Common Linear Classifier Probing Alain and Bengio (2016) is not applicable here since there are no labels with which to train such a classifier. Nevertheless, we attempted a series of embedding visualizations using t-SNE, as detailed in Appendix D and have two noteworthy observations.

The left figure in Fig. 6 shows that the standard deviation of gating factors in each layer increases with higher layers. This aligned with our intuition that the top layers generally require bigger changes to adapt to different tasks, while the bottom layers are generally more shared across tasks.

The second observation, however, is surprising. We compared the t-SNE embedding of Aligner and LLaMA-Adapter trained on both the Alpaca and MetaMath datasets to see their relationship (right of Figure 6). One may guess that the embeddings should be more similar based on tasks since the Alpaca task and math tasks look so different, and Aligner can improve Alpaca's task dramatically, whereas not much for math reasoning relative to LLaMA-Adapter. Rather, the embeddings are much closer to each other for the same method, so much so that the t-SNE position basically overlaps. We then did further comparison over the embeddings, and surprisingly found that approximately half of the numbers in the embeddings are *exactly the same* for both Aligner and LLaMA-Adapter, and many of the rest have very minimal differences. More can be seen in the Appendix D. This shows that it takes very little change for a model to adapt to different tasks. This finding should shed some light on how LLMs work internally.

Additionally, one may wonder how the globally-prefixed Aligner token is related to the locally-prefixed LLaMA-Adapter, such as if it is approximately an average (in the center). We found that this is not necessarily the case as we see in the right half of Figure 6. Other comparisons, such as

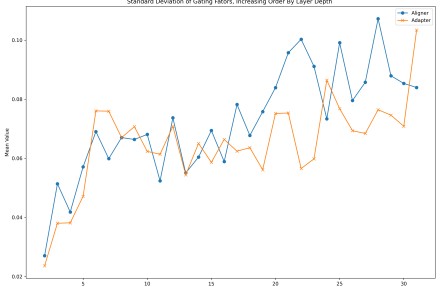 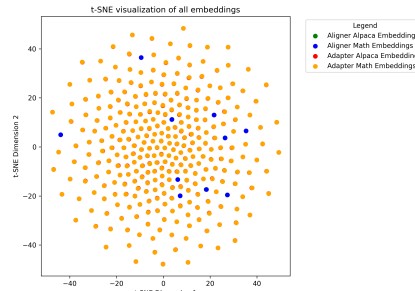

Figure 6: (Left) The standard deviation of gating factors for a layer in Aligner (Blue) or LLaMA-Adapter Model (Yellow). (Right) The t-SNE visualization when putting both Aligner and LLaMA-Adpater embeddings trained on both Alpaca and MetaMath dataset (in total 4 different models' embeddings) altogether.The embedding difference between datasets for the same method comparing to the difference between methods for the same datset is so small that the former one almost completely overlap with each other.

comparing Aligner 1 to Aligner 10, also did not produce meaningful results. More information and experiments can be found in Appendix D.

## 5  DISCUSSION

### 5.1  THEORETICAL ANALYSIS

We contend that our method offers compelling theoretical evidence to substantiate the idea that "forms" or "values" operate orthogonally to "knowledge" and "reasoning" within LLMs.

In traditional linear matrix settings, techniques like eigen decomposition or Singular Value Decomposition (SVD) can yield a matrix factorization in which one factor, typically a sparse one, may be interpreted as representing "form" while the others embody "knowledge". Such decompositions are possible in bilinear (matrix) models (Tenenbaum and Freeman, 1996) and in higher-order, multilinear (tensor) models (Vasilescu and Terzopoulos, 2007) fitted to training data. However, in the context of large neural network models, the feasibility of such a separation remains ambiguous. While the success of various PEFT methods, in view of their rapid learning rates and minimal parameter requirements, strongly suggests that such an orthogonal separation should exist, these methods offer no direct evidence. This is because they rely on dedicated parameters affecting localized modifications. For example, LoRA modifies each linear layer with its own set of parameters, while Prefix Tuning and its variations, such as LLaMA Adapters, introduce layer-specific tunable tokens. Consequently, it remains uncertain whether this "form" truly operates in an orthogonal space.

The most persuasive evidence of orthogonal separation in LLMs would be to achieve it while maintaining comparable performance to traditional methods. Although we cannot perform a linear matrix decomposition, achieving something that aligns with the essence of such a decomposition—a sparse component that globally impacts the remaining components—should be viewed as a practical realization of this separation if the outcomes are equally effective. Our Aligner method seems to fulfill these criteria, thus providing compelling support of our hypothesis.

Moreover, using only a single token essentially rules out the possibility of retaining any knowledge. This stands in contrast to previous methods like LoRA and LLaMA Adapters, which, although smaller than the base models, still involve millions of parameters—enough to encapsulate some degree of knowledge. Such a scale compromises the ability of these methods to serve as unequivocal evidence supporting our hypothesis. The further evidence provided by our third experiment, which effectively tuned a 7 billion parameter model using merely about 100 parameters, substantially strengthens our argument. If an orthogonal "form" component did not exist within LLMs, it would be difficult to offer an alternative rationale for our method's success.

Another intriguing aspect worthy of consideration is that when using only one token in Aligner, the attention weighting effectively becomes nullified as the weight will always be 1 due to the softmax partition function. This implies that the hidden embeddings of the original sequence are essentially being shifted by a constant bias, derived from $P^\top$, and linearly adjusted by $W_V^l$ for each layer. From

a geometric standpoint, assuming that the original sequence embeddings lie on a high-dimensional manifold at each layer, this constant bias acts as a translational shift on that manifold. If we envision the hidden embeddings across layers as a trajectory of movements, alignment is essentially the application of a translational shift along this trajectory. This interpretation aligns well with our intuitive understanding of what "alignment" means in daily language: adapting to a different "way".

## 5.2 APPLICATIONS AND IMPACTS

Our Aligner method is compatible with other PEFT methods such as LoRA and various prefix token approaches. Given the extreme efficiency of our approach, it has the potential to reduce the parameter count in more complex tasks that involve both the acquisition of new knowledge and form adaptation. As mentioned in Section 3.2, some industrial applications can benefit significantly.

In the context of neural architecture design, our method could inspire research into the inclusion of global components. Interestingly, analogous structures exist in the human brain. For example, regions like the Ventromedial Prefrontal Cortex (vmPFC) and Dorsolateral Prefrontal Cortex (dlPFC) (Gazzaniga et al., 2019), which are crucial for moral and value judgments, resemble a global component that interfaces with multiple other brain regions.

Also, Aligner can be used as a probing method to understand if a task is more one of form alignment or reasoning/knowledge improvement. For example, initially, it was unclear if value alignment tasks were mainly about form alignment, but Aligner, using just one token, achieved comparable performance, confirming its role in form alignment. By contrast, for math reasoning tasks, Aligner could not match the performance of state-of-the-art methods without equivalent parameter counts, indicating that math differs from form alignment. This approach can also be applied to less obvious tasks; for example, why pretrained LLMs work well for tasks like molecule generation or classification Qian et al. (2023) is not fully understood. Using Aligner in this context may help reveal how much LLMs achieve it through form alignment or through knowledge acquisition.

Moreover, we posit that our method holds significant promise for advancing AI safety. As AI models grow more powerful, their "black box" nature raises control and safety concerns. Our approach, which enables the encoding of value preferences into a minimal set of tokens, offers a pathway to decoupling the value orientation of an LLM from its functional capabilities, thereby facilitating greater control. Actually, this prospect was a primary motivator of our research. Future work should explore the application of our method to achieving more reliable and controllable AI alignment.

## 5.3 LIMITATIONS

We have not definitively established the capacity of a single token to encapsulate form information. While our SFT experiments with the Alpaca dataset indicate that Aligner with one token is less capable than LoRA, the performance difference is small. It is unclear if the gap is attributable to the lack of hyper-parameter tuning or other random factors. Even though Aligner with one token is inferior, it is also unclear what it failed to learn. The value alignment tasks also did not show clear incompetence. They include failure cases, but such is also the case for LoRA. RLHF (Wang et al., 2023) in real world practice often assimilates millions of training samples, therefore the failures may be attributable to training and data shortcomings. This leaves open the question of scalability with larger datasets, a topic worthy of future exploration.

## 6 CONCLUSIONS

We have introduced a novel Parameter-Efficient Fine-Tuning approach that, using just one or ten tokens, achieves across LLM model scales performances comparable to the state-of-the-art mega-parameter methods like LLaMA-Adapter and LoRA for form alignment tasks. Furthermore, we demonstrated the efficacy of a globally-connected token in alignment tasks, albeit no special advantage over reasoning tasks, which therefore validates the hypothesis that LLMs handles form orthogally to reasoning in the way that form globally affects the way of reasoning process inside LLMs. Our findings inspire significant potential applications and provides insights into the internal mechanisms of large language models, thereby opening up promising avenues for future research.

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
