# OpenReview forum: "Aligner: One Global Token is Worth Millions of Parameters When Aligning LLMs"
_ICLR.cc/2024/Conference — Submitted to ICLR 2024_

### Official Review · Reviewer_o5YH · 2023-10-31

**Soundness:** 2 fair
**Presentation:** 2 fair
**Contribution:** 2 fair
**Rating:** 5
**Confidence:** 4

**Summary:**

This paper proposes a simple modification of the LLama Adapter V1 in order to achieve more parameter-efficient fine-tuning. The novelty is to have one global set of tokens (or in the extreme: only one token) that all layers attend to, rather than per-layer tokens. Attention weights for each layer are separate. The result is a very small set of parameters to fine-tune.
This method (based on Llama2 7B and 13B) is compared against LLama Adapter and LoRA on the Vicuna benchmark for instruction following and the PKU-Beaver dataset for DPO alignment with human values. The results show that the success of the method varies between model sizes and with different numbers of training epochs, but overall scores win ratios of around 0.4-0.5 on Vicuna, and performs better or on par for most categories of the PKU-Beaver set.

While the results seem promising, they lack deeper analysis and a more concise interpretation (see below). Given the small novelty of the technical solution, the paper should have allocated more space for an investigation of the implications of this small change.

**Strengths:**

- The proposed method is attractive for its simplicity and efficiency. It is a simple modification of existing technique, but according to the results effective in reducing parameters significantly while maintaining quality.
- The evaluation is conducted on two benchmarks which represent two important uses of adapters, so it is relatively expressive.

**Weaknesses:**

- In three places it is argued that intuitive inspection confirms the success of the proposed method, by showing one example (Sections 5.1, 5.2, 5.3). The argument is weak, as it’s not a systematic inspection and could have been a cherry-picked example, and is left for subjective evaluation by the reader without any elaboration of why the quality is sufficient. In Section 5.3 this is particularly irritating since quantitative results for the one-token approach are missing, and the reader is left to trust that this example is representative. In 5.1 the example is used to dismiss the quality of the Vicuna evaluations (“too much variance”), which could have easily been tested and quantitatively verified by running the evaluation multiple times and reporting mean and variance.
- The interpretation of the win ratio for Aligner 1 (1 token) in Table 1 is given as “comparable” with the baselines, but it is actually <0.5, so technically worse than the baselines. This interpretation seems rather too optimistic, it is missing an analysis and a focus on the loss cases, because in practice no one wants to deploy a technique that worsens previous results, perhaps even if there’s an win in parameter efficiency. The trade-off between quality and number of parameters for the new and the standard adapter would have been helpful to compare directly in a plot.
- Large portions of the paper are dedicated to reviewing the background, the basics of Llama Adapter and RLHF. Considering that the proposed method is a minor modification, more space should have been rather invested for more exhaustive ablations and deeper analysis of why the method works, how it differs qualitatively from Llama Adapter, and how the hypothesis of separation of form and knowledge can be empirically verified.

**Questions:**

- Why does the 7B model perform better than the 13B model in Table 1? Any hypotheses?
- Is the code open-sourced or published with the paper?

---

> ### Author Response · Authors · 2023-11-22
>
> Thank you your valuable feedback. We would like to clarify the following points:
>
> 1. Showing only one qualitative example in the main paper:
>
> Apologies for the confusion. Due to the page limit we show only one example in the main paper, but we've included many more examples in a supplementary appendix.  We've also now added many failure cases and winning cases along with GPT-4'sjudgements.
>
> We also did not cherry pick our examples. The quantum physics question is commonly used, so we chose to present it in our main paper.  In general, the SFT results quality is stable.
>
> For the value alignment experiments, we did a bit of picking, but only to avoid the unsafe content that comes not only from Aligner but also from LoRA, which appears more often. We had already included plenty of examples in the appendix, and we've now added more failure and winning cases. You can peruse our supplementary appendix to get a sense of that. Please let us know if you want us to include additional examples in the main paper and/or more examples in the supplement.
>
> 2. The "comparable" claim for Aligner-1-Token & analysis of the failure case:
>
> We toned down our claim, and we also provided samples of failure cases in an appendix.
>
> 3. "... in practice no one wants to deploy a technique that worsens previous results, perhaps even if there's an win in parameter efficiency."
>
> This isn't always the case. For industrial scenarios where one needs to provide a customized model to every different user (such as assuming a unique character or replying in a customized format), the parameter efficiency is very important when scaling up to tens of thousands or even millions of customized adapters. For Aligner, even with one million adapters, that's just 5 billion extra parameters (around 10G memory), which can still fit into a single 24GB memory GPU along with a 7B model (13G memory) using standard bfloat16 precision without quantization. By contrast, with LoRA or LLaMA-Adapters, this will consume terabytes with millions of adapters and it'll be necessary to offload to secondary storage, which would drastically slow down the inference time. So, our method could be very useful in saving costs. However, thank you for raising this concern as we must expand on this point in our paper. We agree that our "paper should've allocated more space for an investigation of the implications of this small change".
>
> 4. Sec 5.3 lacking quantitative results for the extended experiments using a single head:
>
> Thank you for your criticism of this section, which we've deleted for the reasons mentioned in Item 3 of our above common comment.
>
> 5. Concern over "how the hypothesis of separation of form and knowledge can be empirically verified":
>
> We believe that we've already largely verified it: The very structure of Aligner -- a minimally sized and globally orthogonal component, along with the fact that its efficacy is on par with other methods, confirms that form functions orthogonally to reasoning/knowledge in LLMs.
>
> However, we do agree that we can further validate our hypothesis. We can also provide a negative example; i.e., for reasoning tasks, Aligner should not exhibit a similar advantage over other higher-parameter and non-global component tuning methods. Since math serves as the most representative reasoning task, we trained them on a math dataset (MetaMath) and tested their performance on a standard benchmark, GSM8K. Aligner indeed does not show a parameter advantage over other methods, thereby validating the hypothesis.
>
> Please see Item 1 in our above common comment and Section 4.3 of our revised paper.
>
> 6. "The trade-off between quality and number of parameters for the new and the standard adapter would have been helpful to compare directly in a plot":
>
> Thank you for your suggestion. In the added math experiments, we included a series of different sizes and showed their performance and size trade off in Fig 5 of the paper.
>
> 7. Our code will be open sourced after we tidy it up, but we can release a version quicker if this is deemed necessary.
>
> 8. Why does the 7B model perform better than the 13B model in Table 1? Any hypotheses?
>
> This may be due to the variation of GPT 4 evaluation and/or due to the training dynamics of the model as a result of hyper-parameter choices. We didn't expend substantial effort to find optimal hyper-parameters, but they definitely matter to a certain extent, which may account for the slightly better performance.

---

> > ### Comment · Reviewer_o5YH · 2023-11-22
> > **Response**
> >
> > Thank you for your clarifications!

---

### Official Review · Reviewer_aiF1 · 2023-10-31

**Soundness:** 1 poor
**Presentation:** 2 fair
**Contribution:** 2 fair
**Rating:** 3
**Confidence:** 4

**Summary:**

This work presents a new parameter-efficient fine-tuning method, which is best described as a reduced version of LLaMA-Adapter, with tuned key+value hidden states, shared across layers with layer-specific gates.

**Strengths:**

- The method is clearly described and simply motivated.

**Weaknesses:**

- The key weakness of this work is that all model evaluation is performed using only model-based evaluation (and specifically model-based comparisons), and not on any labeled benchmark tasks. For a work that is primarily a model-adaptation method, and one that is far lower capacity than full fine-tuning. Moreover, model-based evaluation has been shown to have several limitation in the literature, and I do not believe the field is yet ready to accept results based only on model-based evaluation. For this reason, despite its technical contributions, and I do not think this work can be accepted in its current state.
- Several grammatical/language errors and inconsistencies e.g. "Llama-Adapter" vs "LLaMA-Adapter"
- Strictly speaking, LLaMA-Adapter (and likewise Aligner) do not prepend prefix tokens. They employ a side-channel of attention over a separate set of tokens. The softmax is compute separately from the actual input tokens (see: Eqn 8), and position encodings are not applied to the LLaMA-Adapter/Aligner keys.
- The experiment demonstrating the efficacy of single-head Aligner is a simply a handful of examples of generated outputs. This does not meet the bar of rigorous evaluation for a conference paper.

**Questions:**

- How does this method perform on standard benchmark tasks?
- How do we determine the effective capacity of this method (e.g. on what tasks or in what settings does it underperform/not underperform full-finetuning, or higher-capacity parameter-efficient fine-tuning methods?)

---

> ### Author Response · Authors · 2023-11-22
>
> Thank you for your valuable feedback.
>
> We have added some standard benchmark results to address your questions and concerns. However, we would like to offer the following clarifications:
>
> 1. The rationale for using GPT as the judge:
>
> First, this was the default evaluation method in the papers introducing both the Alpaca and Beaver datasets on which we trained Aligner. It was therefore natural to adopt their methods.
>
> Second, to our knowledge, standard datasets like MMLU (general knowledge) or GSM8K (math) have precise correct answers and only care if the final answer is correct. They generally test the *reasoning ability* of a model rather than the *form* of its output.  By contrast, for alignment tasks like Instruction Following and Value Alignment, which are the focus of our paper, the model outputs are open-ended and there is no single right answer. The "form" of the answer is of importance. So, in our case, the gold standard is human judgment. This is also why the default benchmark methods provided by Alpaca and Beaver are both model-based evaluations, since, despite its shortcomings, thus far a powerful model is the closest surrogate to human judgment for such tasks.
>
> However, we agree that it would help in evaluating Aligner more completely if we include some standard benchmarks, but the purpose this can serve would be different. Rather than directly evaluating the "form" alignment task, it helps to see how our Aligner method affects the reasoning ability of the model in comparison to other adaptation methods, and to further test the hypothesis that form functions orthogonally from reasoning or knowledge. This experiment may also help better validate our hypothesis that form functions orthogonally to reasoning/knowledge in LLMs.
>
> 2. The standard benchmark results:
>
> First, we added the MMLU evaluation that tests general knowledge without training over its dataset, but using the model finetuned on the Alpaca dataset only, along with the raw LLaMA2 base model's performance. The Alpaca dataset does help reasoning ability to some extent since it contains answers that have step by step reasoning and even some math and coding questions. However, it's primarily aiming to tune the model to follow instructions. Therefore, if form functions orthogonally to reasoning, Aligner should perform worse but there should not be a big gap. The results validate our hypothesis.
>
> The evaluation results:
>
> | Alpaca Finetuned Only |  | MMLU Accuracy |  |
> | --- | --- | --- | --- |
> | 7B | Raw Base Model | 3588/14042 | 0.2555 |
> |  | LLaMA-Adapter | 5037/14042  | 0.3587 |
> |  | LoRA | 4958/14042  | 0.3531 |
> |  | Aligner 1 | 4025/14042  | 0.2866 |
> |  | Aligner 10 | 4613/14042  | 0.3285 |
> | 13B | Raw Base Model | 4112/14042 | 0.2928 |
> |  | LLaMA-Adapter | 5056/14042 | 0.3601 |
> |  | LoRA | 5268/14042 | 0.3751 |
> |  | Aligner 1 | 4607/14042  | 0.3281 |
> |  | Aligner 10 | 4835/14042  | 0.3443  |
>
> Second, more importantly, we trained different methods over the MetaMath dataset (a math dataset) on the 7B model and evaluated on GSM8K. This was to assess how much reasoning improvement different methods can attain along with their different parameter sizes. If form functions orthogonally to reasoning in LLMs, Aligner should not have an advantage over other methods, unlike in form alignment tasks (Instruction following and Value alignment).  The results validate our hypothesis. Aligner only catches up the performance of LLaMA-Adapter when they have a similar number of parameters.
>
> | Method  | GMS8K |
> | --- | --- |
> | Aligner 1 (Param 5k)| 70/1319 0.053 |
> | Aligner 10  (Param 42k)| 136/1319 0.1031 |
> | Aligner 100  (Param 410k)| 215/1319 0.1630 |
> | Aligner 300 (Param 1.2M)| 346/1319 0.2623 |
> | LLaMA-Adapter (Param 1.2M)| 334/1319 0.2532 |
> | LoRA (Param 4.2M)| 469/1319 0.3556 |
>
> If the above rationale and plan are unsatisfactory and you have something else in mind, please let us know what specific evaluation(s) you would like us to perform.
>
> 3. Regarding the single-head Aligner experiments:
>
> Thank you for your criticism of this section. We have deleted the section for the reasons explained in Item 3 in our above common comment.
>
> 4. You are right that the LLaMA-Adapter and Aligner is not strictly a prefix token method. We have changed the wording to be more rigorous in the Related Work section.

---

### Official Review · Reviewer_tff3 · 2023-11-02

**Soundness:** 3 good
**Presentation:** 2 fair
**Contribution:** 1 poor
**Rating:** 5
**Confidence:** 3

**Summary:**

This work presents a variant of prefix-token based task alignment method for large language model. Basic idea of the prior LLaMA-adapter is to employ a special short prefix token sequence which are summarized via self-attention, then, linearly combined with the input self-attention part. This work further extend the idea by sharing tokens in each layer, but employing layer specific attention parameters. Experiments on Vicuna benchmark shows comparable performance against LLaMA adapter and LoRA with significantly lower number of parameters for adaptation.

**Strengths:**

- A simple extension to LLaMA-adapter with significantly lower number of parameters, but rivaling the performance with other adapting method.

**Weaknesses:**

- There exist no analysis on the learned parameters especially for tokens. It would be better to quantify the gains by investigating what was learned in the small number of parameters. Also, this work should investigate what is learned by the weight parameters $\beta$.

**Questions:**

See the weaknesses.

---

> ### Author Response · Authors · 2023-11-22
>
> Thank you for your valuable feedback.
>
> We once plotted the distribution of the parameter values of the token, but it is essentially Gaussian and provides no useful insights.
>
> Unfortunately, other common methods like linear classifier probing \cite{linearProbe} are also inapplicable: it requires not only numerous datapoints but also labels for them in order to train this classifier. In our case, we obtain only 1 set of tokens from training over the entire dataset.
>
> The only meaningful approach seems to be to compare the Aligner token with the LLaMA-Adapter tokens using t-SNE or directly. This could potentially reveal how our globally-shared token relates to the local tokens. We have conducted several comparisons and added the results to the main paper and the supplementary appendix. Here's a list of findings:
> 1. The Aligner 1 Token embedding is not necessarily the "average" of Aligner 10 Token or the LLaMA-Adapter, which has 300 tokens.
> 2. When we train the model with a different dataset, the embeddings of Aligner and those of the LLaMA-Adapter remain very close to each other. In fact, approximately half of the numbers in the embeddings are *exactly the same*, and the majority of the rest are still very close. This shows that very little change is needed to dramatically adapt the behaviors of an LLM.
> 3. The standard deviation of the gating factors in a layer is increasing as the layer goes to top, for both Aligner and LLaMA-Adapter. This may be in align with our general intuition that top layers generally requires larger adaptations.
>
> The figures are shown in section 4.4, figure 6 in the main paper and Appendix D in the supplement. We also put some analysis that doesn't yield any finding in Appendix D that you can take a look at. We would be pleased to perform any other suitable data analysis that you can recommend to us.

---

### Author Response · Authors · 2023-11-22
**Summary of the major changes and improvements in response to the reviewers' feedback**

We thank all three reviewers for their valuable feedback. We have made major revisions to the paper, which are highlighted in red text, including the following:

1. Based on the feedbacks from Reviewer 2 and 3, we added some reasoning task evaluations to further validate the hypothesis that "form" functions orthogonally to "reasoning" inside LLMs in Section 4.3. Particularly, we finetuned different methods over the MetaMath dataset (a math dataset) on the 7B model and evaluated on GSM8K. This was to assess how much reasoning improvement different methods can attain along with their different parameter sizes. If form functions orthogonally to reasoning in LLMs, Aligner should not have an advantage over other methods, unlike in form alignment tasks (Instruction following and Value alignment).  The results validate our hypothesis. Aligner only catches up to the performance of LLaMA-Adapter when they have a similar number of parameters.

| Method  | GMS8K |
| --- | --- |
| Aligner 1 (Param 5k)| 70/1319 0.053 |
| Aligner 10  (Param 42k)| 136/1319 0.1031 |
| Aligner 100  (Param 410k)| 215/1319 0.1630 |
| Aligner 300 (Param 1.2M)| 346/1319 0.2623 |
| LLaMA-Adapter (Param 1.2M)| 334/1319 0.2532 |
| LoRA (Param 4.2M)| 469/1319 0.3556 |

2. Based on the feedback from Reviewer 1, we have added embedding visualization in Section 4.4 and in Appendix D to help better understand Aligner.

3. We have deleted the single head variation of Aligner.  Since this exploration used only around 100 parameters, we previously simply wanted to use the instruction following behavior as further evidence that form functions differently from reasoning/knowledge. Our goal wasn't to achieve impressive performance since our point can still be made despite consistently worse performance, so we didn't include a quantitative evaluation.

Since Reviewers 2 and 3 requested quantitative results, we thought of comparing against the raw base model. Even without the Alpaca tuning, the raw base model can still answer questions given the Alpaca dataset's prompt template, despite quality fluctuations. We had erroneously assumed that the base model can't answer questions and attributed the question answering pattern to our single head method, but we discovered that the single head method does not improve the model at all. So we have now deleted that entire section.
We greatly appreciate the reviewer feedback, which prompted us to recognize our erroneous thinking.

4. Based on the feedback from Reviewer 3, we emphasized the potential impact of our method by mentioning possible use cases such as using it as a probing method to understand the nature of a task to be more of form alignment or knowledge acquisition, and also highlighted how the parameter efficiency could benefit industrial applications such as serving massive quantities of customized models.

5. We updated the Vicuna Benchmark evaluation for Instruction Following in Section 4.1. Previously, we had an extra comparison between Aligner 10 Token 8th epoch and other methods for the 13B model. Now we extended training for all of the models to compare the best versions. But only the 13B LLaMA-Adapters has different score, and the change doesn't the conclusion.

We hope that our revisions adequately address the concerns. We believe that our work offers an extremely parameter-efficient method for alignment tasks that is useful in many scenarios, and that our theoretical finding that form functions orthogonally to reasoning within LLMs is of value to the community. Thus, we hope the reviewers might consider increasing their ratings, but, regardless, we are grateful for their feedback, which has strengthened our paper.

Our ensuing comments will address the specific points raised in each review.

---

### Meta-Review · Area_Chair_Jj4f · 2023-12-05

**Metareview:**

This paper introduces a PEFT method for aligning billion-scale LLMs. Reviewers and AC appreciate the simplicity of the proposal method. However, all reviewers consistently have concerns on the quality of the paper. AC thanks the authors for providing a rebuttal, which unfortunately does not change the initial opinions of the reviewers. In particular, reviewers are concerned about diverse aspects, ranging from writing to experiments and analysis. AC hopes the reviews are helpful for the authors to prepare for a future version of the paper.

**Justification For Why Not Higher Score:**

All reviewers consistently have concerns on the quality of the paper and vote for rejection. The paper is a clear rejection according to the reviewers' scores and comments.

**Justification For Why Not Lower Score:**

N/A

---

### Decision · Program_Chairs · 2024-01-16

Reject